# Adaptive Hybrid Storage Format for Sparse Matrix–Vector Multiplication on Multi-Core SIMD CPUs

**Shizhao Chen** [1,2], **Jianbin Fang** [2,*] , **Chuanfu Xu** [1,2] **and Zheng Wang** [3]

1    Institute for Quantum Information & State Key Laboratory of High Performance Computing, National University of Defense Technology, Changsha 410073, China
2    College of Computer Science and Technology, National University of Defense Technology, Changsha 410073, China
3    School of Computing, University of Leeds, Leeds LS2 9JT, UK
*    Correspondence: j.fang@nudt.edu.cn

**Abstract:** Optimizing sparse matrix–vector multiplication (SpMV) is challenging due to the non-uniform distribution of the non-zero elements of the sparse matrix. The best-performing SpMV format changes depending on the input matrix and the underlying architecture, and there is no "one-size-fit-for-all" format. A hybrid scheme combining multiple SpMV storage formats allows one to choose an appropriate format to use for the target matrix and hardware. However, existing hybrid approaches are inadequate for utilizing the SIMD cores of modern multi-core CPUs with SIMDs, and it remains unclear how to best mix different SpMV formats for a given matrix. This paper presents a new hybrid storage format for sparse matrices, specifically targeting multi-core CPUs with SIMDs. Our approach partitions the target sparse matrix into two segmentations based on the regularities of the memory access pattern, where each segmentation is stored in a format suitable for its memory access patterns. Unlike prior hybrid storage schemes that rely on the user to determine the data partition among storage formats, we employ machine learning to build a predictive model to automatically determine the partition threshold on a per matrix basis. Our predictive model is first trained off line, and the trained model can be applied to any new, unseen sparse matrix. We apply our approach to 956 matrices and evaluate its performance on three distinct multi-core CPU platforms: a 72-core Intel Knights Landing (KNL) CPU, a 128-core AMD EPYC CPU, and a 64-core Phytium ARMv8 CPU. Experimental results show that our hybrid scheme, combined with the predictive model, outperforms the best-performing alternative by 2.9%, 17.5% and 16% on average on KNL, AMD, and Phytium, respectively.

**Keywords:** SpMV; performance; machine learning; sparse matrix format



## 1. Introduction

Sparse matrix–vector multiplication (SpMV) (SpMV is defined as $\mathbf{y} = \mathbf{A} \cdot \mathbf{x}$, where $\mathbf{A}$ is an input *sparse* matrix, and the input vector $\mathbf{x}$ and the output vector $\mathbf{y}$ are dense) is a common computing block for many scientific applications and deep learning workloads [1,2], and is often responsible for the performance bottleneck of these applications [3]. As the multi-core design is becoming the mainstream architecture, it is important to ensure that SpMV is well optimized for the underlying multi-core architecture.

A key challenge for optimizing SpMV is to reduce the irregular memory accesses resulted from the uneven distribution of non-zero elements in the sparse matrix. Prior work in this area applies a set of techniques such as blocking [4–6] and data rearrangement [7–9] to improve the data locality and load balancing. As a consequence, there is an intensive body of work on designing new matrix storage formats by reorganizing the non-zero entries of the sparse matrices [4,5,7–10]. Unfortunately, studies have shown that there is no "one-size-fit-for-all" SpMV storage format, and the optimal format changes depending on the input matrix and the underlying architecture [11,12].

To best match the SpMV optimization to the underlying hardware, we would like to bring the best of different SpMV storage schemes and have a scheme to adaptively choose between multiple SpMV storage formats for the underlying hardware on a per input base. To this end, a promising solution is to use machine learning to adaptively select an optimal storage format for a given matrix and processor. Recent work has employed machine learning to dynamically select from a set of SpMV storage formats for a given matrix [13–16]. However, such an approach requires the programmer to either provide different implementation variants of the program to use different storage formats or ensure that the overhead of runtime storage format conversion can be amortized by the benefit of format selection [17,18].

As an alternative, other works [4,6,11,19] developed a hybrid solution by combining two native SpMV storage formats in a single framework. For example, the hybrid (HYB) format [4] works by storing the typical number of non-zeros per row in the ELLPACK (ELL) format [20] and the remaining entries of exceptionally long rows in the COO format. Its implementation computes a histogram of the row sizes and determines the largest number K such that using K columns per row in the ELL portion meets a certain objective measure. In the follow-up work, Guo et al. present ECV-HYB [11], a hybrid scheme designed for GPU executions. This approach first sorts the rows of the matrix based on the number of non-zero elements of the row. It then partitions the rows into short and long rows, where short rows are stored in the ELL format, and long rows are stored in the vectorized compressed sparse row (CSR) format [10]. The benefit of doing so is to increase coalescing memory access for a wider range of non-zero patterns, which thus improves memory access performance on GPUs.

The core idea of a hybrid scheme is to partition the matrix into two portions: non-zero elements that lead to regular memory accesses can be processed by a SIMD-friendly SpMV kernel, and the other part with an irregular memory access pattern is processed by an SpMV kernel suitable for irregular sparsity patterns [19]. Thus, these hybrid approaches are able to use the best of both native storage formats, while avoiding maintaining various storage formats. However, the prior hybrid approaches are targeted for GPUs, and their native formats are not suitable for multi-core CPUs, which has close to one hundred SIMD cores and distributed on-chip cache slices. In particular, the COO format used by many hybrid schemes [4,19] is unsuitable for SIMD units and can fail to handle the irregular part efficiently. Furthermore, a key tuning parameter for these hybrid approaches is the partitioning threshold, or *K*, for determining the position to split the matrix. As we will show later in the paper, choosing the right *K* is important for achieving good performance, but the optimal settings vary from one matrix to another (see Section 3). Our work is designed to address these two drawbacks.

In this paper, we present a new hybrid sparse matrix storage format, namely HYB5, that is optimized for multi-core CPU processors. To better exploit the SIMD feature of the multi-core design and balance SpMV workloads across rows, our approach upgrades HYB [4] by employing two latest SpMV storage formats, the SELL-C-$\sigma$ format [9] and the CSR5 format [8]. To store matrix elements in the two storage formats, we first break columns of the input matrix into two partitions in the vertical direction, where the left part exhibits regular memory access patterns, and the right part has various row lengths and irregular memory accesses. We then store the left part with regular memory accesses in the SELL-C-$\sigma$ format, and the remaining in the CSR5 format, where each format has its own computation kernel for SpMV operations.

As a departure from prior work [4,11] that relies on the users and their analytical models to determine the right threshold, *K*, for matrix partitioning, we develop an adaptive approach by using machine learning to train a predictive model to determine the optimal *K* on a per input basis automatically. The model takes as input a set of quantifiable attributes or features from the input matrix and predicts the optimal value for *K*. The model is first trained offline using a set of training sparse matrices. The trained model can then be applied to any new, unseen sparse matrices to predict the best value to use for *K*. Our approach frees

the programmer from manually tuning the optimal storage configuration. As it is often difficult to anticipate what sparse matrices will be encountered, our automatic approach thus provides a better generalization ability by automatically choosing the right threshold.

We evaluate our approach by applying it to 956 sparse matrices on 3 multi-core CPU platforms: a 72-core Intel Knights Landing (KNL) CPU processor, a 128-core AMD EPYC processor, and a 64-core Phytium ARMv8 processor. We compare our approach against five universal formats, including CSR [10], CSR5 [8], ELL [20], SELL-C-$\sigma$ [9] and HYB [4]. We also compare our approach with the state-of-the-art, best-performing format, CVR [21], which is specifically tuned for the KNL architecture. On KNL, our approach outperforms all competing approaches, with an average improvement of up to 96.7%. On AMD EPYC, our approach outperforms all alternative schemes with an average improvement of up to 25.5%. On Phytium, our approach outperforms all competing approaches by an average improvement of up to 70%. We show that our predictive model is highly accurate in choosing the partitioning factor, delivering 98.3%, 97.0% and 91.0% of the best available performance given by a theoretically perfect predictor on KNL, EPYC, and Phytium, respectively.

This paper provides the following techniques contributions:

- We propose a new hybrid storage format (HYB5) for sparse matrices, which relies on a mixture of SELL-C-$\sigma$ and CSR5 and their individual SpMV implementations (Section 4).
- We use a machine learning based approach to automatically predict the optimal matrix segmentation threshold (Section 5).
- We demonstrate that, by using the best of both worlds, our approach can achieve a better performance than the state-of-the-art SpMV implementations on three representative multi-core CPU architectures (Section 6).

## 2. Background

### 2.1. Sparse Matrix–Vector Multiplication

Sparse matrix–vector multiplication (SpMV) is of the form $\mathbf{y} \leftarrow \mathbf{A} \cdot \mathbf{x}$, where $\mathbf{A}$ ($M \times N$) is a sparse matrix, and $\mathbf{x}$ ($N \times 1$) and $\mathbf{y}$ ($M \times 1$) are dense vectors. Figure 1 gives a simple example of SpMV, where the number of non-zero elements (*nnz*) of $\mathbf{A}$ is 8 and $M = N = 4$.

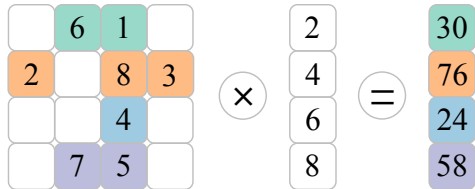

**Figure 1.** A SpMV example with a $4 \times 4$ matrix and a vector.

SpMV is a numerical algorithm based on sparse matrix. The use of compressed storage formats reduces memory footprints, but it also increases the number of discontinuous memory accesses and increases the memory access overhead. Since the compressed storage destroys the temporal and spatial locality in the calculation, it is difficult to optimize the SpMV on the cache-based CPU. To improve the computational efficiency of SpMV, the existing implementation needs to be improved by combining the characteristics of the target architecture and matrix storage format with the characteristics of the SpMV algorithm itself.

### 2.2. Sparse Matrix Storage Format

Due to a large number of zeros existing in sparse matrices, using dense matrix formats will lead to a waste of storage and computing resources. Thus, researchers have invented various sparse matrix storage formats, i.e., merely storing the non-zero elements and their indices of a matrix.

**COO** The *coordinate* (COO) format (i.e., IJV format) is a particularly simple storage scheme. The arrays `row`, `col`, and `data` are used to store the row indices, column indices, and values of the non-zero elements. This format is a general sparse matrix representation because the required storage is always proportional to the number of non-zero elements for any sparsity pattern. Different from other formats, COO stores explicitly both row indices and column indices. Table 1 shows an example matrix in the COO format.

**Table 1.** Matrix storage formats and their data structures for the sparse matrix shown in Figure 1.

| Representation | Specific Values |
|---|---|
| COO | $row = [0, 0, 1, 1, 1, 2, 3, 3]$ <br> $col = [1, 2, 0, 2, 3, 2, 1, 2]$ <br> $data = [6, 1, 2, 8, 3, 4, 7, 5]$ |
| CSR | $ptr = [0, 2, 5, 6, 8]$ <br> $indices = [1, 2, 0, 2, 3, 2, 1, 2]$ <br> $data = [6, 1, 2, 8, 3, 4, 7, 5]$ |
| CSR5 | $ptr = [0, 2, 5, 6, 8]\ tile\_ptr = [0, 1, 4]$ <br> $tile\_des : bit\_flag = [T, T, F, F\|T, T, T, F],$ <br> $y\_off = [0, 1\|0, 2], seg\_off = [0, 0\|0, 0]$ <br> $indices = [1, 0, 2, 2\|3, 1, 2, 2]$ <br> $data = [6, 2, 1, 8\|3, 7, 4, 5]$ |
| ELL | $data = \begin{bmatrix} 6 & 1 & * \\ 2 & 8 & 3 \\ 4 & * & * \\ 7 & 5 & * \end{bmatrix}\quad indices = \begin{bmatrix} 1 & 2 & * \\ 0 & 2 & 3 \\ 2 & * & * \\ 1 & 2 & * \end{bmatrix}$ |
| SELL | $data = \begin{bmatrix} 6 & 1 & * \\ 2 & 8 & 3 \\ 4 & * \\ 7 & 5 \end{bmatrix}\quad indices = \begin{bmatrix} 1 & 2 & * \\ 0 & 2 & 3 \\ 2 & * \\ 1 & 2 \end{bmatrix}$ <br> $slices = [3, 2]$ |
| SELL-C-$\sigma$ | $data = \begin{bmatrix} 2 & 8 & 3 \\ 6 & 1 & * \\ 7 & 5 \\ 4 & * \end{bmatrix}\quad indices = \begin{bmatrix} 0 & 2 & 3 \\ 1 & 2 & * \\ 1 & 2 \\ 2 & * \end{bmatrix}$ <br> $slices = [3, 2]$ |
| HYB | $ELL{:}data = \begin{bmatrix} 6 & 1 \\ 2 & 8 \\ 4 & * \\ 7 & 5 \end{bmatrix}\quad indices = \begin{bmatrix} 1 & 2 \\ 0 & 2 \\ 2 & * \\ 1 & 2 \end{bmatrix}$ <br> $COO{:}\ row = [1], col = [3], data = [3]$ |
| CVR | $data = \begin{bmatrix} 6 & 2 \\ 1 & 8 \\ 4 & 3 \\ 7 & 5 \end{bmatrix}\quad column\_indices = \begin{bmatrix} 1 & 0 \\ 2 & 2 \\ 2 & 3 \\ 1 & 2 \end{bmatrix}$ |

**CSR** The *compressed sparse row* (CSR) format is the most popular general-purpose sparse matrix representation. This format explicitly stores column indices and non-zero elements in array `indices` and `data`, and uses a third array `ptr` to store the starting nonzero index of each row in the sparse matrix (i.e., row pointers). For an $M \times N$ matrix, `ptr` is sized of $M + 1$ and stores the offset into the $i^{th}$ row in `ptr[i]`. Thus, the last entry of `ptr` is the total number of non-zero elements. Table 1 illustrates an example matrix represented in CSR. We see that the CSR format is a natural extension of the COO format by using a compressed scheme. In this way, CSR can reduce the storage requirement. More importantly, the introduced `ptr` facilitates a fast query of matrix values and other interesting quantities, such as the number of non-zero elements in a particular row.

**CSR5** To achieve near-optimal load balance for matrices with any sparsity structures, CSR5 first evenly partitions all nonzero entries to multiple 2D tiles of the same size. Thus, when executing parallel SpMV operation, a compute core can consume one or more 2D tiles, and each SIMD lane of the core can deal with one column of a tile. Then the main skeleton of the CSR5 format is simply a group of 2D tiles. The CSR5 format has two tuning parameters, $\omega$ and $\sigma$, where $\omega$ is a tile's width and $\sigma$ is its height, which is set to the size of the SIMD execution unit and on-chip memory strategy of the processor, respectively. CSR5 is an extension to the CSR format [8]. Apart from the three data structures from CSR, CSR5 introduces another two data structures: a tile pointer `tile_ptr` and a tile descriptor `tile_des`. Table 1 illustrates an example matrix represented in CSR5, where $\omega = \sigma = 2$.

**ELL** The `ELLPACK` (ELL) format is suitable for the vector architectures [20]. For an $M \times N$ matrix with a maximum of **K** non-zero elements per row, ELL stores the sparse matrix in a dense $M \times K$ array (`data`), where the rows with fewer than **K** are padded. Another data structure, `indices`, stores the column indices and is zero-padded in the same way as that of `data`. Table 1 shows the ELL representation of the example sparse matrix, where **K** = 3 and the data structures are padded with `*`. The ELL format would waste a sizable amount of storage. To mitigate this issue, we can combine ELL with another general-purpose format, such as CSR or COO.

**SELL and SELL-C-$\sigma$** *Sliced* ELL (SELL) is an extension of the ELL format by partitioning the input matrix into strips of **C** adjacent rows [22]. Each strip is stored in the ELL format, and the number of non-zero elements stored in ELL may differ over strips. Thus, a data structure `slice` is used to keep the strip information. Table 1 demonstrates a matrix represented in the SELL format when **C** = 2. A variant to SELL is the SELL-C-$\sigma$ format, which introduces sorting to save storage overhead [9]. That is, they choose to sort the matrix rows not globally but within $\sigma$ consecutive rows. Typically, the sorting scope $\sigma$ is selected to be a multiple of **C**. The effect of local sorting is shown in Table 1 with **C** = 2 and $\sigma = 4$.

**HYB** The HYB format [11] is a combination of ELL and COO, and it stores the majority of matrix non-zero elements in ELL and the remaining entries in COO [4]. Typically, HYB stores the *typical* number of non-zero elements per row in the ELL format and the *exceptionally* long rows in the COO format. In the general case, this *typical* number (**K**) can be calculated directly from the input matrix. Table 1 shows an example matrix in this hybrid format with **K** = 2.

**CVR** The CVR format is an SpMV representation targeting the efficient vectorization developed especially for the Intel KNL processor [21]. The number of columns in CVR formats equals the number of SIMD lanes. It stores the matrix switching from row order to column order, while trying to fill the "data" with non-zero elements. Both the format conversion and the SpMV implementation are SIMD friendly. Table 1 shows an example matrix in CVR format, with two SIMD lanes.

### 3. Motivation

This section presents the motivation of our work and outlines the design choices of our approach.

**Why using a hybrid format?** Studies have shown that no single matrix storage format performs best across matrix datasets and/or architectures [11,12]. As a result, work has been proposed to build an analytical model to pick an optimal storage format for a given dataset and architecture [18]. However, such an approach requires managing various storage formats in a single program, and the overhead of format conversion could be non-negligible during the runtime. As an alternative, a hybrid storage format works by mixing two (or more) native formats in a single framework. Recent works [4,11] have demonstrated the promising results of a hybrid scheme. Our work builds upon these recent works by proposing a new hybrid scheme to target the emerging multi-core architectures.

**Why combine SELL-C-$\sigma$ and CSR5?** As we will show later in Section 6, CSR5 [8] and SELL-C-$\sigma$ [9] are two well-performing sparse matrix storage format on multi-core architectures. Figure 2 shows the performance measurements of the two storage formats on KNL when applying the respective SpMV kernel to 956 matrices. Here, the x-axis shows the regularity (denoted as `variation`) of the input matrix. The variation is calculated by computing the standard deviation of non-zero elements on the row dimension of the input matrix. Therefore, the larger the variation is, the more irregular the matrix will be. Overall, we see that CSR5 (marked with red triangles) performs better than SELL-C-$\sigma$ (marked with blue plus) for irregular matrices (i.e., when `variation` is large), while SELL-C-$\sigma$ performs better for the regular matrices (i.e., when `variation` is small). It can be observed from Figure 2 that CSR5 works better when `variation` is large. Therefore, we expect that a hybrid storage format combining these native formats would exploit the advantages of both worlds by storing the left few non-zero elements per row in SELL-C-$\sigma$ and the remaining (possibly irregular) non-zero elements in CSR5. Figure 3 shows that our approach, namely HYB5 (SELL-C-$\sigma$+CSR5), significantly outperforms HYB (ELL+COO) on KNL when using a right-partitioning parameter *K*.

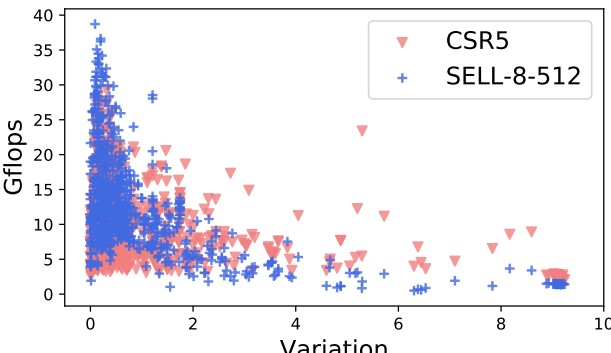

**Figure 2.** SpMV performance of SELL-8-512 and CSR5 for SpMV performs on matrices with different irregularities (variations). The larger the variation is, the more irregularity a matrix has.

**Why is choosing the right K important?** Figure 3 shows how the SpMV performance changes over the partition parameter K for the `cage12` and `Cube_Coup_dt0` matrix from the SuiteSparse matrix collection [23] on Intel KNL (Section 6.1). We note that using a right K gives significant performance benefit over SELL-C-$\sigma$ and CSR5. However, an inappropriate K setting can also hurt performance seriously, as shown by the point marked with **A**. Thus, selecting a suitable partitioning parameter K is of great significance for the SpMV performance. Therefore, to unlock the potential of our hybrid scheme, we also develop a scheme to dynamically determine the right K on a per matrix basis.

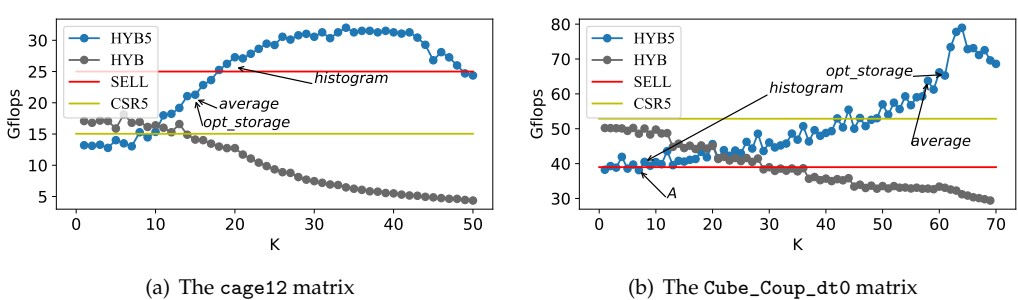

(a) The `cage12` matrix                                              (b) The `Cube_Coup_dt0` matrix

**Figure 3.** How the performance of HYB and HYB5 changes as the partitioning threshold K changes on two sparse matrices (**a**,**b**).

Figure 3 also shows the performance of the prior analytical models: `average` [12], `histogram` [4], and `opt_storage` [19]. We note that the `histogram` model outperforms the

average and `opt_storage` one for `cage12`, while their performance is the other way around for `Cube_Coup_dt0`. For both cases, the obtained SpMV performance via the analytical models is not the best. This is due to the fact that the selection of K depends not only on the input matrix features, but also on the underlying architectures and native storage formats. These first two models can only capture the input matrix features. The `opt_storage` threshold selection is based on minimizing the memory footprint of the HYB matrix format. However, the SpMV performance trends for HYB and HYB5 are different as the threshold changes as shown in Figure 3. Therefore, we need a model that is capable of taking the input matrix features, the underlying architectures and native storage formats into account. In this work, we will use machine learning techniques to build such a model.

## 4. Our Approach

### 4.1. HYB5 Overview

HYB5 is a hybrid sparse matrix storage scheme that combines SELL-C-$\sigma$ [9] and CSR5 [8]. With HYB5, an input matrix is partitioned into two portions in the vertical direction: the left regular portion is stored in the SELL-C-$\sigma$ format, and the right irregular portion is stored in the CSR5 format. Figure 4 shows how a sparse matrix A of size $16 \times 16$ with 76 nonzero elements is represented in the HYB5 format. In this example, K is set to be 3, and matrix A is first divided into two sub-matrices of the same size as A. The left sub-matrix is stored in the SELL-C-$\sigma$ format. Due to the fact that the right portion becomes even more irregular than it was, it is necessary to compress this part to reduce the processing overhead and save storage space.Thus, the right sub-matrix is compressed first, then the rows without non-zero entries are removed (i.e., *empty rows*) and stored in the CSR5 [8] format.

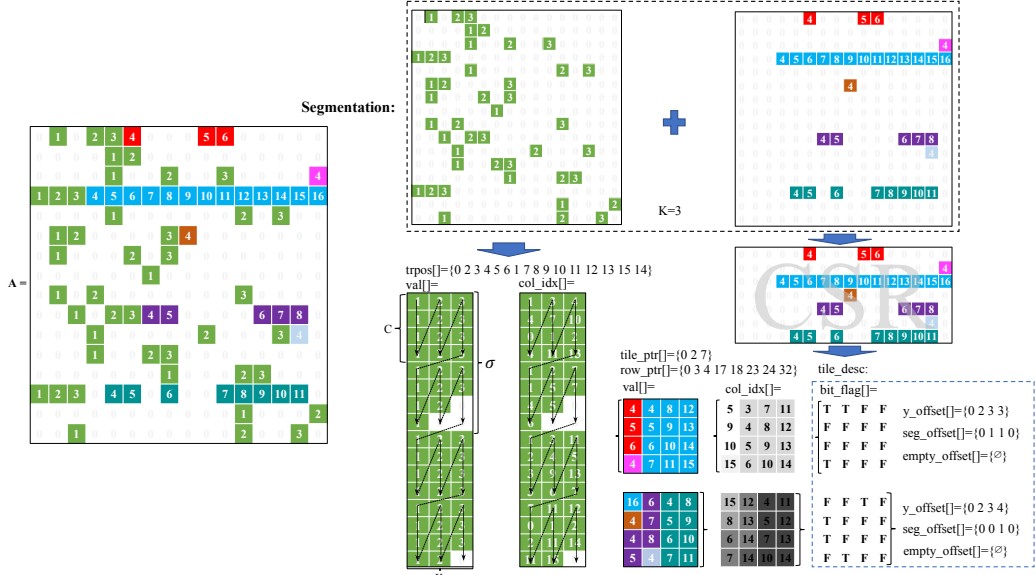

**Figure 4.** Represent a sparse matrix A with the HYB5 format. How a sparse matrix A is represented in HYB5 and SELL-C-$\sigma$. In this example, matrix A is of $16 \times 16$ size and has 76 non-zeros. Colored cells denote non-zero elements, and white cells are zeros. The arrows indicate the memory accessing order.

Compared to the native SELL-C-$\sigma$ format, HYB5 can further reduce the storage and processing overhead. As an illustrative example in Figure 5, the same matrix A is represented in SELL-4-8. We see that there are fewer padding zeros when K = 3 compared to K = 8 with the same configurations. Then, the remaining sparse rows with various lengths are handled by the CSR5 format. By doing so, HYB5 can make use of the best of both worlds. Note that HYB5 is not a simple combination of the two native formats, and we have to choose the right partitioning position between the two portions on a per matrix basis (Section 4.2).

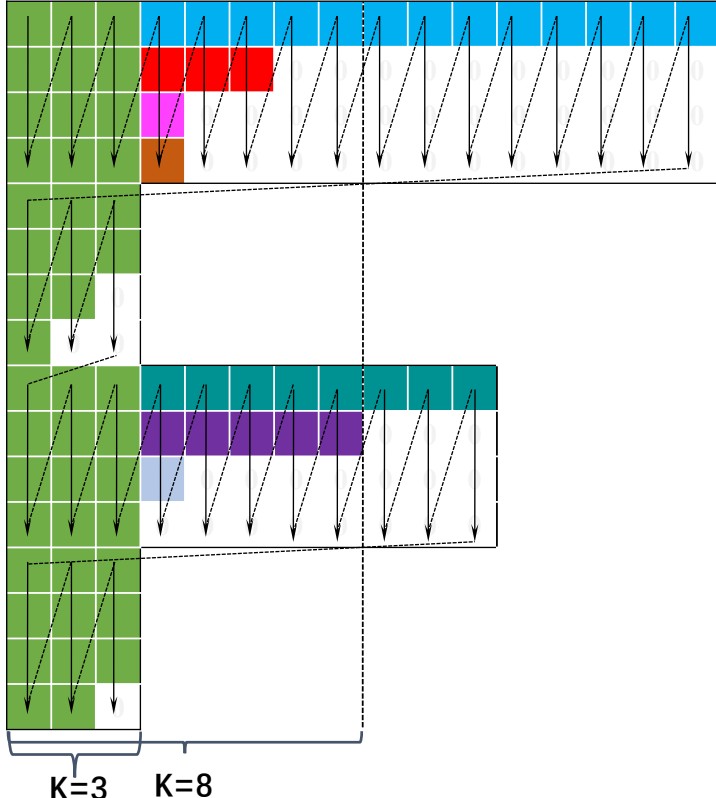

**Figure 5.** Represent the same matrix A with the SELL-4-8 format.

### 4.2. Matrix Segmentation

The key to representing a sparse matrix in the HYB5 format is to determine how to partition the matrix. As a first step, the input matrix A has to be divided into two sub-matrices, where their number of rows and columns are equal to those of matrix A. As shown in Figure 4, the size of the sub-matrices is 16 × 16. During matrix partitioning, we only have to save non-zero elements for both sub-matrices. The divided non-zero elements remain in their original locations for each sub-matrix. As we have mentioned, a partitioning parameter K is used to determine the position where to cut a row of the matrix. The first K non-zero elements are stored in the left sub-matrix. If there are fewer than K non-zeros, then all non-zero elements in this row are stored in the left sub-matrix. In Figure 4, K = 3 and the first three non-zero elements of each row are represented in SELL-C-$\sigma$. The remaining non-zero elements within this row are stored in the right sub-matrix in CSR5. The resultant HYB5 structures are outlined in the upper right-hand portion of Figure 4.

As illustrated in Algorithm 1, for each row, we use an array `row_counter` to record the number of non-zero elements stored in row `coo_row_index[i]`, i.e., the row index of the *i*th non-zero element. When the non-zero elements of this row are accumulated to K, the rest are stored into the right matrix. In other words, the non-zero elements before K of each row will be stored as the left matrix and represented in the SELL-C-$\sigma$ format. The right sub-matrix will be first converted into the CSR format and then to the CSR5 format. We expect that there are as few padding zeros as possible in the left sub-matrix, letting CSR5 handle the very long non-zero rows.

### 4.3. Compressed CSR

Before being converted into CSR5 format, the right submatrix needs to be converted to the CSR format. Figure 4 shows that the right sub-matrix outlined has much fewer non-zero elements comparing with the original matrix **A**, resulting in many *empty rows* with a large processing overhead. Specifically, array `row_ptr`, which is used by both CSR and CSR5 format, contains many duplicate elements. Taking the right submatrix as an ex-

ample, $row\_ptr = \{0 \quad 0 \quad 3 \quad 4 \quad 4 \quad 17 \quad 17 \quad 17 \quad 17 \quad 18 \quad 23 \quad 23 \quad 23 \quad 24 \quad 24 \quad 24 \quad 32\}$. Besides, buffer `empty_offset` (stores the correct positions of empty rows used when computing the partial sum) of the CSR5 also contains duplicate entries. Therefore, we compress the matrix to only retain the rows with non-zero elements.

---

**Algorithm 1** Convert COO to HYB5

---

1: **function** CONVERTTOHYB5(*coo_mtx*∗, **k**)
2:     **p** ← 0
3:     **q** ← 0
4:     **for** $i = 0$ to **nnz do**
5:        **if** $row\_counter[coo\_row\_index[i]] >= K$ **then**
6:           $csr\_data\_stru[p] \leftarrow coo\_data\_stru[i]$
7:           $row\_counter[coo\_row\_index[i]] + +$
8:           $p + +$
9:        **else**
10:          $sell\_data\_stru[q] \leftarrow coo\_data\_stru[i]$
11:          $row\_counter[coo\_row\_index[i]] + +$
12:          $q + +$
13:        **end if**
14:     **end for**
15:     $CCSR\_mtx* \leftarrow$ CompressCSR(*csr_data_stru*)
16:     $CSR5\_mtx* \leftarrow$ ConvertToCSR5(*CCSR_mtx*)
17:     $SELL\_mtx* \leftarrow$ ConvertToSELL(*sell_data_stru*)
18: **end function**

---

During the conversion process, we introduce a new data structure (`nnz_row_idx`) to record the original index of each row. For the CSR format, we need to change `row_ptr` to eliminate the entries of indicating empty rows. If two adjacent elements in `row_ptr` are equal, we regard that there exists an empty row. Algorithm 2 shows that we remove the empty rows, convert `row_ptr` to `com_row_ptr`, and record the original index of the remaining rows in `nnz_row_idx`. Figure 4 shows that the size of the right submatrix is significantly reduced by compressing the raw data. Consequently, the size of array `row_ptr` is reduced, and it is unnecessary to use `empty_offset`. When storing the result to **y** which stores the results of SpMV (e.g., y = Ax), we use `nnz_row_idx` to retrieve the original row indices for the corresponding non-zeros. Note that we have not changed the column index of the non-zero elements. By doing so, we guarantee that the SpMV results are stored to the right locations of vector **y**.

---

**Algorithm 2** Compress CSR format.

---

1: **function** COMCSR(*row_ptr*∗, *com_row_ptr*∗, *nnz_row_idx*∗)
2:     $j = 0$
3:     $p = 0$
4:     **for** $i = 1$ to $N$ **do**
5:        **if** $row\_ptr[i] != row\_ptr[i-1]$ **then**
6:           $j + +$
7:           $com\_row\_ptr[j] \leftarrow row\_ptr[i]$
8:           $nnz\_row\_idx[p] \leftarrow i - 1$
9:           $p + +$
10:        **end if**
11:     **end for**
12: **end function**

---

### 4.4. Format Conversion

We split the sparse matrix, A, into two sub-matrices, which are initially stored in the COO format. We then convert the left sub-matrix to the SELL-4-8 format, as shown

in Figure 4. Here arrays `val` and `col_idx` are transposed from the row-major order to the column-major order. After sorting the rows of the matrix by the number of non-zero elements, we use array `trpos` in the SELL-C-$\sigma$ format to the original row indices. We set $C = 8$ for Intel Knights Landing with 512-bit SIMD units, $C = 4$ for both AMD EPYC and Phytium 2000+. To reduce zeros entries, we set the sorting scope $\sigma = 8$. As indicated by the arrow in Figure 4, the matrix values and column indices are accessed in the column-major order. The SELL-C-$\sigma$ format divides value and column index into $N/C$ blocks, with each having $C \times K$ elements.

The right compressed sub-matrix in CSR can be directly converted into the CSR5 format. As shown in Figure 4, we partition the non-zero entries to two $4 \times 4$ tiles. When performing SpMV operations, a compute core can deal with multiple tiles, and each SIMD lane of the core can work on a column of a tile [8]. For each tile, CSR5 introduces a tile pointer `tile_ptr`, a tile descriptor `tile_desc` and reuses the three CSR data structures (`value`, `row_ptr`, and `col_idx`). Similar to the SELL-C-$\sigma$ format, the `val` data and the `col_idx` data are transposed from the row-major order to the column-major order.

## 5. Adaptive Parameter Tuning

This section introduces how to build a model for selecting a right position of cutting a sparse matrix.

The model estimates the performance gain for a given K value for the target matrix. The predictive model is used as a utility function to quickly search for the optimal K. Predictions are made based on a set of numerical values, or feature values, of the target matrix as described in Section 5.2.

Our predictive model is built upon the scikit-learn machine learning package [24]. We use a support vector machine (SVM) based regression model with an `rbf` kernel, which is effective in modeling non-linear data. An alternative to our regression-based approach is to build a classifier to directly predict the optimal *K*. However, a classifier can only choose a value that has been seen in the training data, due to the nature of classification algorithms. We avoid this drawback by employing an unconstrained based approach—our regression-based model can be used for arbitrary K values (even those that were not presented during training) because the model takes the value as an input to estimate the potential gain.

Building and using such a predictive model follows the 3-step process for supervised machine learning: ❶ generate training data, ❷ train a predicted model, and ❸ use the predictor.

### 5.1. Training the Predictor

The offline training only needs to be performed once for a given architecture. To target a new architecture, we need to generate and label training samples for the targeting hardware. This is done by profiling the benchmarks under different K to find the optimal K for each training program. However, the process of feature selection, training and model selection is fully automated and can remain unchanged for a new architecture.

Figure 6 depicts the process of training the predictive model. To collect training data, we profile each training matrix under a given K to find the best-performing K. We normalize the measured SpMV running time against the default setting, $K_{average}$, that computes K by counting the average number of non-zero elements per row. For each training matrix, we also extract the feature values.

#### 5.1.1. Generating Training Data

We use cross-validation to train and test our approach. We use 80% of the sparse matrices from the SuiteSparse matrix collection [23] to train our model and then test the trained model on the remaining matrices. For each training matrix, we store the sparse matrix using our hybrid format under a given K. We then profile the resulting SpMV performance to find the best-performing K value.

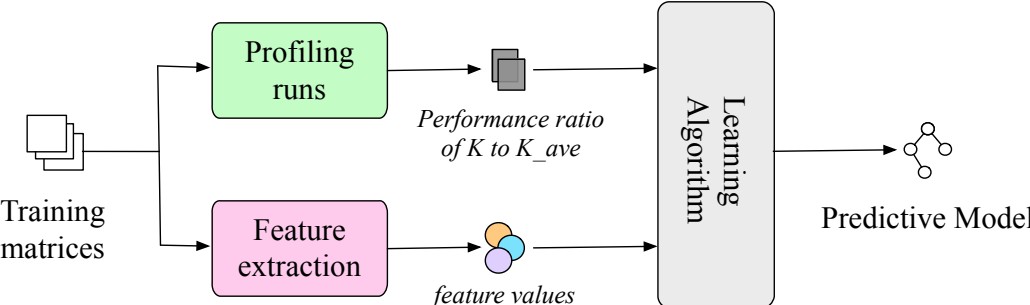

**Figure 6.** The training process of our predictive model.

As the range of possible values that K can take is too large, it is prohibitively expensive to exhaustively search for the optimal K for every training matrix. Instead, we sample 10% of our training matrices and find that the best K often stays around $K_{average}$ and $K_{histogram}$, where $K_{histogram}$ is the number given by a histogram algorithm similar to HYB. As a result, for each training matrix, we profile the resulting SpMV performance of around $K_{average}$ or $K_{histogram}$, [$K_{average}$-30, $K_{average}$+30] and [$K_{histogram}$-30, $K_{histogram}$+30], where the lower bound is capped at 0. To generate training data, we measure the resulting SpMV performance for each K value for a given training matrix and record the best-performing value. On average, we profile 30–120 different values of K. To minimize the impact of profiling noise, we profile each K value setting multiple times until the upper and lower confidence bounds are smaller than 5% under a 95% confidence interval setting. As part of the training data collection, we also collect a set of feature values (Section 5.2) for each training matrix.

### 5.1.2. Building the Model

The performance ratio as a label of the sample, along with their corresponding feature set, is passed to our supervised learning algorithm. The learning algorithm tries to find a correlation between the feature values and optimal representation labels. The output of our learning algorithm is a version of our SVM-based model. To relate the target models to architectures and matrix features, the training processor is performed for distinct platforms separately. Since training is performed off-line and only needs to be carried out once for a given architecture, this is a one-off cost.

### 5.1.3. Training Overhead

The total training time of our model is comprised of gathering the training data and then building the model. Gathering the SpMV performance data of the formats consumed most of the total training time. It took around two days for our platforms. In contrast, gathering features of matrices and building the model took a negligible amount of time, less than 10 s and 10 ms, respectively.

### *5.2. Feature Engineering*

The key to building a successful predictor is developing the right features to characterize the input. Our predictive model is based exclusively on static, hardware-agnostic features of the target matrix, and no dynamic profiling is required. The features are extracted using our Python script. Since the models are trained separately for our platforms, the hardware-specific features are implicitly embedded in the models.

### 5.2.1. Feature Selection

Table 2 presents a full list of all our selected features. $K_{average}$ and $K_{histogram}$ are obtained by a traditional method presented in [4]. The time overhead to obtain the features of one matrix is in the order of milliseconds. As we have 47,956 groups of training samples from 756 matrices, we would like to reduce the number of features. To do so, we first combine some of the individual features. Through feature combination, we reduce the

number of features to 6, which are *nnz_frac*, *n_rows*, *nnz_std*, *variation*, $K_{average}$, $K_{histogram}$. We apply the wrapper method [25,26] to further obtain the best subset of features and then confirm *nnz_frac*, *n_rows*, *nnz_std*, *variation* and $K_{average}$ as our features. The importance is shown in Table 2. The evaluation criteria for the subset of features are the predictive accuracy of the model. The wrapper method means that the model is trained by using different subsets of features until it is accurate enough or all the subsets of features have been tested.

**Table 2.** Features used in our predictive model.

| Features | Description | Importance |
|---|---|---|
| *n_rows* | number of rows | 33.71% |
| *n_cols* | number of columns | |
| variation | matrix regularity | 3.96% |
| *nnz_frac* | percentage of non-zero elements | 42.72% |
| *nnz_tot* | total number of non-zero elements | |
| $K_{histogram}$ | K is obtained by a histogram algorithm | |
| $K_{average}$ | average number of non-zero elements per row | 2.10% |
| nnz_min | minimum number of non-zero elements per row | |
| *nnz_max* | maximum number of non-zero elements per row | |
| nnz_std | standard derivation of non-zero elements per row | 17.51% |

### 5.2.2. Feature Scaling

In the final step, we scale each of the extracted feature values to a common range (between 0 and 1) to prevent the range of any single feature being a factor in its importance. We record the minimum and maximum values of each feature in the training dataset in order to scale the feature values of an unseen matrix. We also clip a feature value to make sure it is within the expected range during deployment.

### 5.3. Runtime Deployment

Deployment of our predictive model is designed to be simple and easy to use. To this end, our approach is implemented as an API. The API has encapsulated all of the inner workings, such as feature extraction and matrix format translation. We also provide a source to source translation tool to obtain the available optimal K for HYB5. With the predicted results, a high-performance HYB5 can be obtained at runtime.

## 6. Evaluation

### 6.1. Experimental Setup

**Hardware Platforms.** We use three multi-core CPUs: a 72-core Intel Knight Landing (KNL) multi-core processor, a 128-core AMD EPYC CPU, and a 64-core Phytium ARMv8 CPU. The KNL processor has a peak performance of 3Tflops for double-precision operations [27]. Each KNL core has an AVX512 unit supporting four threads and running at 1.3 GHz. Each core has a private L1 data and L1 instruction cache of 32 KB. The AMD EPYC CPU features a peak performance of 3.5 Tflops double-precision operation [28,29]. The AMD CPU used has two sockets, each having 64 AVX2 cores and 256 MB L3. Each EPYC core supports two hardware threads running at 3.35 GHz, and a private 32 KB L1 data and a private 64 KB L1 instruction cache. `Phytium 2000+` has 64 high-performance ARMv8 compatible `xiaomi` cores running at 2.2 GHz. The entire chip offers a peak performance of 563.2 Gflops for double-precision operations. The 64 hardware cores are organized into 8 `panels`, where each panel connects a memory control unit. Each core has a private L1 cache of 32KB for data and instructions, and every four cores share a 2 MB L2 cache [30].

**Systems Software.** Our platforms run a customized Linux operating system with a kernel v3.10.0 on KNL, v4.18.0 on AMD EPYC, and v4.19.46 on Phytium 2000+. We use Intel icc v17.0.4 on KNL, gcc v8.3.1 on AMD EPYC, gcc v9.3.0 on Phytium 2000+, with the default "-O3" option. We use the OpenMP threading model with 272 threads on KNL, 256 threads on EPYC and 64 threads on Phytium.

**Datasets.** Our experiments use 956 square matrices from the SuiteSparse matrix collection [23]. The number of nonzero elements of the matrices ranges from 100 K to 200 M. The dataset includes both regular and irregular matrices, covering domains from scientific computing to social networks.

**Native Format Settings.** In the experiments, we use double precision to store the matrices and vectors. On KNL for CSR5, $\omega = 8$, $\sigma = 12$, and for SELL-C-$\sigma$, C = 8. On AMD EPYC for CSR5, $\omega = 4$, $\sigma = 16$, and for SELL-C-$\sigma$, C = 4. On Phytium 2000+ for CSR5, $\omega = 4$, $\sigma = 16$, and for SELL-C-$\sigma$, C = 4. This setting is used in accordance with 512-bit SIMD units on KNL, 256-bit SIMD units on AMD EPYC and 128-bit SIMD units on Phytium. And $\sigma = 512$ on KNL, $\sigma = 256$ on AMD EPYC, and $\sigma = 256$ on Phytium 2000+ for SELL-C-$\sigma$. Accordingly, HYB5 also uses the same configurations for its native formats.

**Evaluation Methodology.** We use cross validation to evaluate our approach by randomly splitting the 965 matrices into two parts: 756 matrices for training and 200 matrices for testing. We learn a model with the training matrices. We then evaluate the learned model by applying it to make a prediction on the 200 testing matrices. We use a regression model, and the performance of the model is based on the accuracy of the predicted results. The performance of the parameter K predicted by the model is injected into HYB5 during runtime. We use the ratio of performance of predicted results to the Oracle performance to represent the accuracy of the model.

*6.2. Compared with State-of-the-Art*

Figure 7 shows the performance comparison of SpMV for six state-of-the-art storage formats on KNL and for five storage formats on Phytium 2000+ and AMD EPYC. On KNL, HYB5 outperforms the state-of-the-art storage formats by 96.7% for HYB, 60.6% for CSR, 25.2% for SELL-C-$\sigma$, 7.4% for CSR5, and 2.9% for CVR on average. On AMD EPYC, HYB5 outperforms the state-of-the-art storage formats by 19.8% for SELL-C-$\sigma$, 21.5% for CSR5, 17.5% for HYB, and 25.5% for CSR on average. On Phytium 2000+, HYB5 outperforms the state-of-the-art storage formats by 70% for CSR5, 32.3% for CSR, 23.3% for SELL-C-$\sigma$ and 16% for HYB on average. Therefore, HYB5 obtains the best performance on the three multi-core CPUs with SIMDs. This is because HYB5 can make the best of both SELL-C-$\sigma$ and CSR5, i.e., it can achieve a good data locality and enable balanced loads for both regular and irregular matrices. We did not test CVR on AMD EPYC and Phytium 2000+, because it can only work on architectures that support the `Intel avx512` intrinsics, which are not supported on AMD EPYC or Phytium 2000+.

Figure 7a shows that the CSR-based SpMV performs bad in most cases on KNL. The main reason is that the code of the CSR-based SpMV is not explicitly vectorized for KNL. Among the six formats, the performance of HYB is the worst. This is due to fact that its native formats (ELL and COO) are not competitive, compared to SELL-C-$\sigma$ and CSR5. Moreover, the selection of the cutting parameter K is not optimal (Section 3).

In Figure 7b, we see that on AMD EPYC, the CSR5-based SpMV performs worse than that on KNL. The main reason is that the CSR5-based SpMV has an intensive usage of the `scatter` operation which is supported by the KNL `avx512` intrinsics. However, AMD EPYC uses `avx2` and has no `scatter` instruction. Thus, we have to manually scatter the result data, which leads to inefficient vectorization. On the other hand, the SELL-C-$\sigma$-based SpMV still performs well. By using the best of both worlds, HYB5 still retains its optimal performance which again indicates HYB5's cross-platform advantages.

Figure 7c shows that, CSR5 performs the worst among all the formats on Phytium 2000+. This is because the CSR5-based SpMV is not explicitly vectorized for Phytium 2000+.

In contrast, the performance of SELL-C-$\sigma$ looks promising even without vectorization. The work [12] states that the performance of vectorized SELL-C-$\sigma$-based SpMV on Phytium 2000+ is limited by the missing support of `gather` and `scatter`. However, our HYB5 format can still take the advantage of CSR5 in that it excels in handling exceptionally long rows. Note that HYB of combining ELL and COO have achieved the second best performance, running after our HYB5 format.

Figure 8 further illustrates the performance of CVR with the HYB5's native formats on KNL. We note that the performance of the SELL-C-$\sigma$ is not satisfying, when the input matrix has a large variation. For such cases, this format pads many zero elements, which leads to a large storing and computing overhead. Among all the single storage formats, CVR yields the most competitive performance with ours on KNL. This is due to the fact that CVR uses a vectorization-friendly data layout, specialized for the KNL vector units. However, we have not chosen CVR as the native format of HYB5, because the CVR performance is sensitive to the regularity of matrices. That is, CVR performs worse than our chosen native formats for matrices with very large or small variations (Figure 8). This is further illustrated by the long tail of the CVR violin in Figure 7a. This also conforms to our motivation of building our new format HYB5.

### 6.3. Compared with Predictive Models

Figure 9 shows the performance comparison between our model and three existing analytical models (`K_average`, `K_histogram` and `K_opt_storage` [19]) on the three target architectures. The y-axis represents the achieved performance of the four K-selection methods in terms of `accuracy` calculated by Equation (1). For each matrix, the SpMV performance with the K obtained from one of the three methods is $Gflops_{k\_test}$, and the SpMV performance with the best K is $Gflops_{k\_best}$.

$$Accuracy = Gflops_{k\_test}/Gflops_{k\_best} \tag{1}$$

We see that, for the three tested platforms, in general, the accuracy of our model is optimal, which is significantly better than that of $K_{histogram}$, $K_{opt\_storage}$ and $K_{average}$. In terms of average accuracy, our model outperforms $K_{histogram}$, $K_{opt\_storage}$ and $K_{average}$ by 26.1%, 5.4%, and 7.7%, respectively, on KNL; 61.1%, 47.3% and 59.4%, respectively, on AMD EPYC; and 72.6%, 109.2% and 77.7%, respectively, on Phytium 2000+. Figure 7c shows that, although the native formats of HYB5 have a poor performance, our model on Phytium 2000+ can provide superior performance and still outperform the state-of-the-art formats. Note that $K_{average}$ yields better performance than $K_{histogram}$ on KNL (Figure 9a).

### 6.4. Compared with Best Individual Formats

We have mentioned that there are two general solutions to the "one-size-fits-all" issue: selecting the best individual format and using the hybrid solution. Figure 10 compares the performance of these two solutions. From Figure 10a,b, we see that, for the small matrices (i.e., with few non-zero elements), the performance of HYB5 is worse than that of the best individual format. However, the performance of HYB5 is competitive for large sparse matrices. Figure 10c also shows that HYB5 performs better than the best individual formats, especially for the large matrices. Overall, we summarize that using the best individual formats generally outperforms using a hybrid solution. However, programmers have to maintain various individual storage formats and endure the converting overhead across formats. By contrast, we have to maintain only one hybrid format during software development. This particularly helps when developing large-scale applications. Therefore, using a hybrid solution wins in maintenance cost, but suffers a loss in performance. It is up to the programmers to make the final choice.

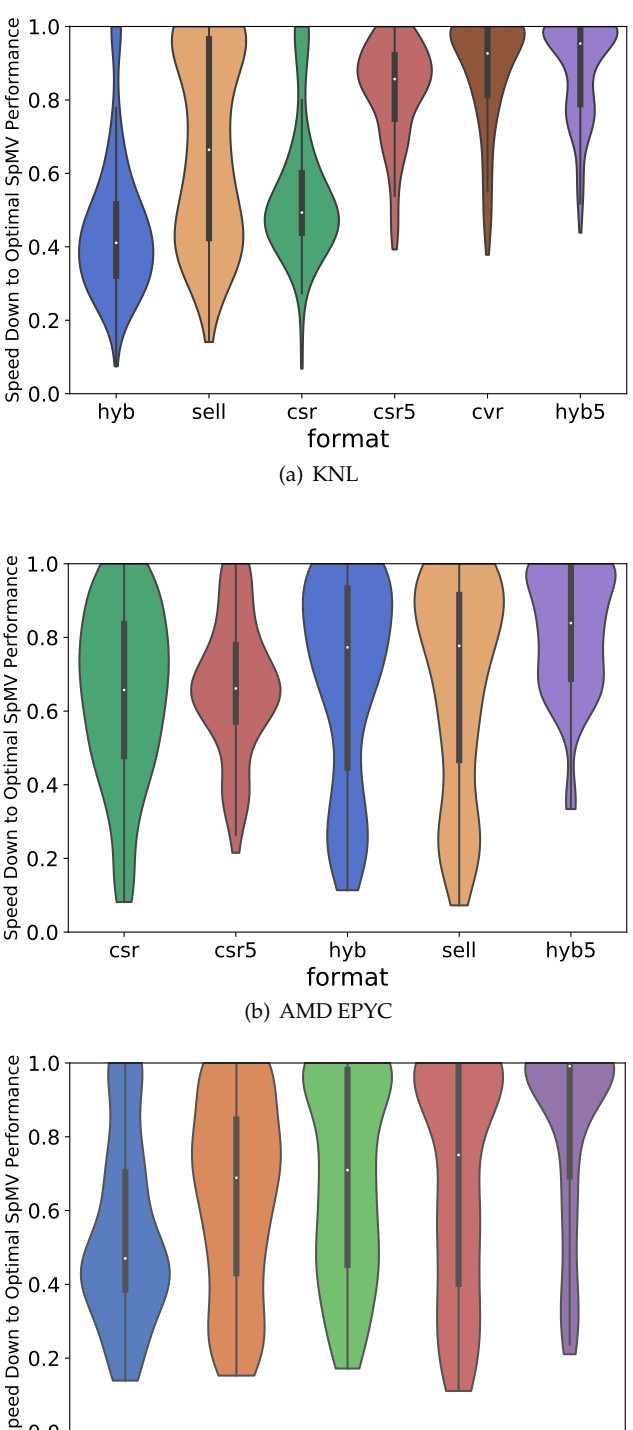

(a) KNL

(b) AMD EPYC

(c) Phytium 2000+

**Figure 7.** Comparing HYB5 performance to state-of-the-art formats. The violin diagrams show the reduction ratio of the compared formats for the relatively optimal format of SpMV performance for each matrix. The shape of the violin represents the distribution of the acceleration ratio over the data set, and the bold black line is the location of 50% of the data set.

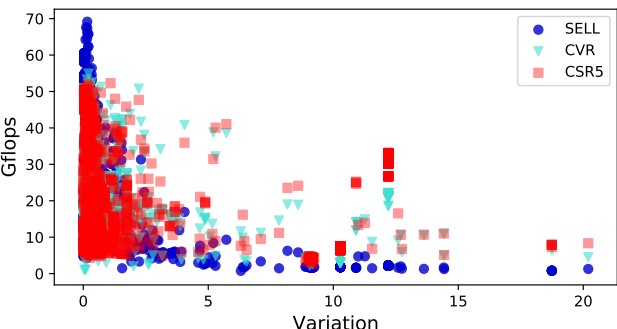

**Figure 8.** SpMV performance of SELL-8-512, CSR5, and CVR with matrices of different variations. The x-axis is variation (`vardata`), which represents the regularity of a given matrix. As `var` increases, the regularity degree of matrices decreases.

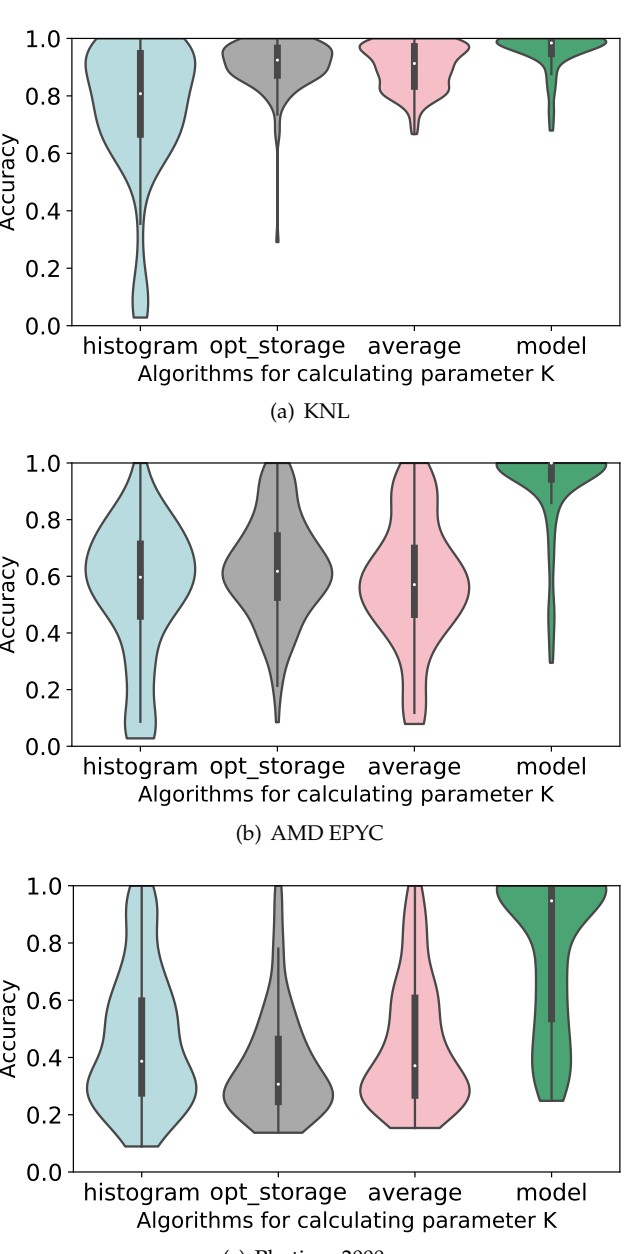

**Figure 9.** The violin diagram shows the comparison of three methods of obtaining parameter K by showing its corresponding SpMV performance on the three target architectures.

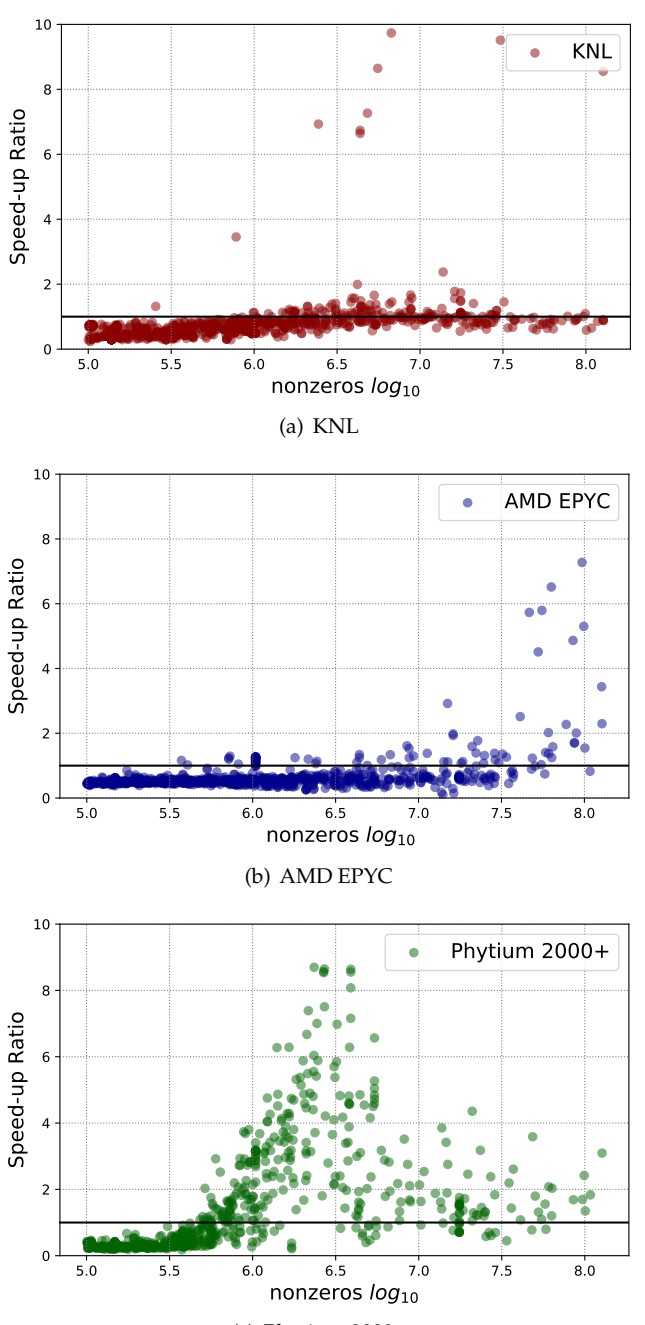

**Figure 10.** The scatter diagrams show the speedup ratio of performance of HYB5-based SpMV with the optimal K and the optimal performance from using the other compared formats for each matrix on three target platforms.

## 7. Related Work

SpMV has been extensively investigated on multi-core CPUs with SIMDs [10,31,32]. Recently, the effort was dedicated to new storage formats and/or adaptive autotuning frameworks for performant and scalable SpMV.

**New sparse matrix storage formats** have been designed to enable efficient SpMV by matching various inputs to multi-core CPUs with SIMDs [3,4,10,33]. Xie et al. propose a sparse matrix representation CVR [21] for efficient vectorization. CVR is insensitive to the irregularity and sparseness of SpMV, so it can handle a large number of scale-free matrices on Intel Knights Landing. Liu et al. proposed CSR5 [8], which is a tile-based format. This format enables high-throughput SpMV on both CPUs and GPUs for both regular

and irregular matrices. The format conversion from CSR to CSR5 is claimed to be a few SpMV operations. On Intel KNC, Liu et al. identified and addressed several performance bottlenecks of SpMV [5]. They exploited the salient architecture features of KNC and used specialized data structures with careful load balancing to obtain satisfactory performance. Moritz et al. proposed a SIMD-friendly format SELL-C-$\sigma$ [9]. This format is a variant of `Sliced ELLPACK` by reordering the rows of a matrix stored in Sliced ELLPACK, which aims to decrease the padding overhead and increase the utilization of vector units. The sorting range is determined by $C$ and $\sigma$. Coronado-Barrientos et al. designed a new format `AXC` [34] to improve the SpMV performance for Intel Xeon Phi. `AXC` improves the SpMV performance by avoiding indirect access to vector x and reducing cache misses. This format can outperform CSR for 12 data sets. Chen et al. proposed the `CSR-SIMD` [35] format to utilize SIMD units of ARMv8-based FT1500A and x86-based Intel Xeon Phi fully. This format compresses the non-zero elements into many variable-length pieces of data with consecutive memory access addresses to improve the accessing locality of vector **x**.

**Adaptive optimization methods** of SpMV based on machine learning techniques have attracted interests in recent years [15,36–41]. This is due to the fact that the SpMV performance is determined by a combination of the storage format, the platform architecture, and the input dataset. For different architectures, there is no single format that can enable consistently better performance for all datasets. It is thus necessary to select the best matrix representation for a given architecture and data input. Chen et al. employed a decision tree to build a predictive model to help choose an optimal matrix representation from five widely-used formats: CSR, CSR5, ELLPACK, SELL-C-$\sigma$, and HYB for FT2000+ and Intel KNL [12]. Then, they analyzed the working process of the decision tree in detail [42]. Cui et al. presented a deep learning mechanism for SpMV format selection best suited for a given data set [39]. They converted the matrices into graphs and directly made them inputs to the neural network eliminating the need for manually selecting features. Zhou et al. designed regression models and neural network-based time series prediction models to capture the influence imposed on the existing SpMV program performance by the overhead and benefits of format prediction and conversions [17].

**Using hybrid storage formats** can leverage the advantages of both formats. HYB is the first hybrid format for SpMV optimization on the GPU [4]. It combines ELLPACK and COO as a new format HYB, which can outperform its native formats. HYB determines the segmentation position of sparse matrices with a histogram algorithm, which is outperformed by our model (Section 6.3). Our work upgrades HYB with two new native storage formats and uses machine learning to select the cutting factor.

## 8. Conclusions

We presented a new hybrid SpMV storage format, HYB5, which is designed to utilize the hardware parallelism provided by modern multi-core CPUs. HYB5 aims to bring the best of CSR5 and SELL-C-$\sigma$. It achieves this by storing non-zero matrix elements with irregular patterns in the CSR5 format and the remaining in the SELL-C-$\sigma$ format. This hybrid scheme allows one to employ an efficient parallel kernel to perform SpMV on regular non-zero elements and use the CSR5 to mitigate the impact of irregular matrix accesses. One of the key challenges of our approach is how to determine the partition between the two storage formats. We achieve this by employing machine learning to automatically construct a predictive model from training samples to predict the right matrix partition on a per matrix basis. The learned model can then be used for any unseen matrix. We evaluate our approach by applying it to over 900 matrices on three distinct multi-core CPU architectures. Experimental results show that our approach outperforms all alternative SpMV storage formats across the evaluation platforms. As the future work, it would be interesting to work with the deep learning method to automatically extract matrix features.

**Author Contributions:** Conceptualization, J.F., S.C. and C.X.; methodology, J.F. and S.C.; software, S.C.; validation, S.C., J.F. and Z.W.; formal analysis, S.C., J.F.; investigation, S.C., J.F., C.X. and W.Z.; resources, J.F., C.X. and W.Z; data curation, S.C. and J.F. All authors have read and agreed to the published version of the manuscript.

**Funding:** The NNW project (program no. TC228S03J).

**Institutional Review Board Statement:** Not applicable.

**Informed Consent Statement:** Not applicable.

**Data Availability Statement:** Not applicable.

**Conflicts of Interest:** The authors declare no conflict of interest.

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
