# Peer review of "Adaptive Hybrid Storage Format for Sparse Matrix–Vector Multiplication on Multi-Core SIMD CPUs"

_applsci, doi:10.3390/app12199812_

Round 1

Reviewer 1 Report

The main content is acceptable.

However, some bit concern about time complexity of the proposed method is important that should be evaluated.

For example, Authors mentioned that " A hybrid scheme combining multiple SpMV storage formats allows 4 one to choose an appropriate format to use for the target matrix and hardware". Combining two algorithms leads to increase time complexity. Please check and discuss on it.

Please add a flowchart to show procedure of the proposed method.

Authors can add some future works on the Conclusion section.

Author Response

We thank the reviewer for their insightful comments.

R1.1 A key tuning parameter for hybrid approaches is the partitioning threshold, or K, for determining the position to split the matrix. We note that in Figure 2, when we choose an inappropriate K setting can hurt the SpMV performance of hybrid formats seriously. We further discuss this issue in Section 4.1.

R1.2 We supplement Algorithm 2 to show the compression process of CSR format. Besides, Algorithm 1 shows the detailed process of converting a COO format matrix to HYB5 format. And figure 3 shows the data structure of HYB5 format in detail. 

R1.3 As the future work, it would be interesting to work with Deep Learning method to automatically extract matrix features. 

Reviewer 2 Report

Summary:

The authors present a hybrid sparse matrix storage format optimized for multi-core CPU processors and use machine learning to automatically predict the optimal matrix segmentation threshold

General concept comments:

This work on a new hybrid SpMV storage format (combination of CSR5 and SELL-C-s) is scientifically sound and well presented. All methods are adequately described and the conclusions are supported by the results. For ease of readability, the authors should consider adding an abbreviation section in the paper that defines the formats mentioned. Additionally, the authors should ensure all abbreviations (especially formats) are defined at first mention (for example, COO format is not defined). The authors should discuss the most contributing features in Table 1 or at least provide a plot on feature importance. Due to these reasons, I recommend accepting the article after the comments are satisfactorily addressed.

Author Response

We thank the reviewer for their insightful comments.

R2.1 We add a Background section to introduce SpMV and the matrix format abbreviations mentioned in this paper.

R2.2 We add a column to Table 2 for the feature importance.